# Structure and Rheological Properties of Glycerol Monolaurate-Induced Organogels: Influence of Hydrocolloids with Different Surface Charge

**DOI:** 10.3390/molecules25215117

**Published:** 2020-11-04

**Authors:** Runan Zhao, Shan Wu, Shilin Liu, Bin Li, Yan Li

**Affiliations:** 1College of Food Science and Technology, Huazhong Agricultural University, Wuhan 430070, China; ZRN951219@163.com (R.Z.); wushan531108@163.com (S.W.); slliu2013@mail.hzau.edu.cn (S.L.); libinfood@mail.hzau.edu.cn (B.L.); 2Key Laboratory of Environment Correlative Dietology (Huazhong Agricultural University), Ministry of Education, Wuhan 430070, China; 3School of Materials and Engineering, Zhengzhou University, Zhengzhou 450001, China; 4Functional Food Engineering &Technology Research Center of Hubei Province, Wuhan 430070, China

**Keywords:** organogels, hydrocolloids, surface charge, thermal-sensitivity, viscoelastic property

## Abstract

Organogel (OG) is a class of semi-solid gel, entrapping organic solvent within a three-dimensional network, which is formed via the self-assembly of organogelators. In the present study, OG was produced by glycerol monolaurate (GML) as organogelator. The influence of hydrocolloids with different surface charges (chitosan (CS), konjac glucomannan (KGM) and sodium alginate (SA)) on the physiochemical properties of OG was investigated. Rheological studies demonstrated that OG and pure hydrocolloid solution showed shear-thinning behavior. After incorporation of the hydrocolloid, the initial viscosity of OG was lowered from ~100 Pa·s to <10 Pa·s, and then the viscosity increased to more than 100 Pa·s at a low shear rate of 0.1–0.2 s^−1^, which subsequently decreased with a higher shear rate. OGs in the presence of hydrocolloids still kept the thermo-sensitivity, while the melting point of the OG decreased with the incorporation of hydrocolloids. Hydrocolloid addition greatly shortened the gelling time of the OG from 21 min to less than 2 min. The presence of hydrocolloids increased the particle size of oil droplets in the molten OG. Some aggregation and coalescence of oil droplets occurred in the presence of positive-charged CS and negative-charged SA, respectively. After gelling, the gel structure converted into a biphasic-like network. Hydrocolloids improved the hardness, stickiness and the oil-holding stability of OGs by 18.8~33.9%. Overall, hydrocolloid incorporation could modulate the properties of OGs through their different surface charge properties. These novel OGs have potential as nutrient carriers or low-fat margarine alternatives and avoid the trans-fatty acid intake.

## 1. Introduction

Organogel (OG) is a kind of semi-solid gel system, which is formed by self-assembling of gelators in organic solvents to form a three-dimensional network structure [1]. This system can use its own physicochemical properties to change the physical structure of liquid oil into a solid network without generating any trans-fatty acids [2,3]. On account of its unique texture, plasticity, physical stability and sensory properties, OGs have been widely used in the food industry, such as cream, mayonnaise, or fat replacer [4,5,6]. Many researchers have reported that the texture and performance of OGs can be regulated by selecting oil phase, gelators or the addition of other ingredients and other processing or preparation methods [7]. Cerqueira et al. found that OGs prepared with long chain triglycerides (LCT) were more easily gelled and had higher gel strength than medium chain triglycerides (MCT), since MCT could rotate the dipole-dipole to affect gel crystallization [8]. Arezou et al. reported that carnauba wax/adipic acid-based OG had better thermal behavior, crystallinity and oil binding ability, which can be attributed to the synergistic interaction between the two molecules to form new intramolecular or intermolecular hydrogen bonds [2]. Similarly, some organogelator mixtures, such as lecithin or γ-oryzanol and β-sitosterol, behenic acid and ethylcellulose, fatty acids and fatty alcohols, monoglycerides and phytosterols, have also shown the synergistic effects [9,10,11]. Sharifi et al. found that high intensity ultrasound treatment can modify olive oil/propolis wax OG, which can induce nucleation by forming small crystals and form a strong network with high oil binding capacity [12]. Gökçe et al. used high-speed homogenization and micro-irradiation to prepare carbopols OGs, which can greatly reduce the preparation time and energy required [13].

Although the preparation method and properties of OGs could be optimized by different strategies, OGs still show poor compatibility with many other food ingredients and food systems, which makes them less consumer friendly [7]. Hydrocolloids are common ingredients in the food system, acting as thickening, gelling agent, texture and others, which would interact with other components in the food system [14]. We hypothesized that hydrocolloids could regulate the performance of OGs, which would broaden the application of OGs in food systems. Hence, we selected three kinds of common hydrocolloids, including chitosan (CS, positive), konjac glucomannan (KGM, neutral) and sodium alginate (SA, negative). OG was produced by using glycerol monolaurate (GML) as the organogelator and medium chain triglyceride (MCT) as the organic phase. GML is a naturally occurring fatty acid molecule and antimicrobial agent, which is also widely available as a homeopathic supplement, and is extensively used as a food preservative and emulsifier [15]. The rheological properties and microstructure of OG were characterized and compared in the absence and presence of hydrocolloids. The present work would be useful to expand the potential applications of OGs to replace trans- or saturated fatty acids in functional food products.

## 2. Results and Discussion

### 2.1. Formation of OG in the Absence and Presence of Hydrocolloids

When a certain amount of organogelator is added to the oil phase, OG can be spontaneously formed by controlling the temperature [16,17]. In the present study, OG was formed by the aggregation of GML at a temperature under its melting point. As shown in Figure 1, GML-induced OG was transparent and thermally reversible, formed by the self-assembly of GML (Figure 1). The texture and viscosity of the OG were affected by the content of GML [18]. OG could not be produced when GML content was less than 3% (data not shown). At a higher temperature, the molten OG could disperse into the aqueous phase, which could be water or hydrocolloid solution. The transparent solution became milky turbid and was converted into gel after cooling, which was named as MOGs (Figure 1). The texture of this gel after mixing with hydrocolloid solution was more uniform than that after mixing with water. We think that the gelling mechanism is still due to the self-assembly of GML, while hydrocolloid molecules would interface this process (Figure 1). The incorporation of hydrocolloid solution altered the gelling time and stability of the OGs. Table 1 shows that the gelling time of MOGs was less than 2 min, and significantly shorter than that of OG (21 min). CSOG and KGMOG had good stability and there was no phase separation after centrifugation. Compared with the OG, the centrifuge stability of CSOG and KGMOG was improved by about 30%, and SA addition enhanced the stability of OG from 65.3% to 84.1%. The results indicated that the stability of the OG was significantly improved by the addition of hydrocolloids. It was reported that the shear stress exerted on the food during food processing was much higher than that of the centrifuge in the laboratory [9]. Therefore, the results demonstrated that OGs containing hydrocolloids could be more resistant against the shearing force during food processing.

Afterwards, we measured the texture of the OGs in the absence and presence of hydrocolloids. Hydrocolloids significantly increased the firmness of the OGs, followed by CS > KGM > SA. Chitosan was the most effective, which indicated the interaction between negative GML and positive CS in the system was the strongest. The addition of hydrocolloids also improved the stickiness of the OGs. The higher stickiness meant that the OG structure would not easily disintegrate, which was related to the strength of the internal gel structure [19]. Adhesiveness is the tendency of food to resist separation from a contacting material. Highly adhesive processed gel food (such as margarine) is not popular with consumers, as it is difficult to separate from the package [20]. Table 1 shows that the adhesiveness of the OG decreased with the addition of hydrocolloids. Among them, CSOG had the lowest adhesiveness, which was 93.74 g·sec. This information indicated that the addition of hydrocolloids could be designed to reduce the adhesiveness of the OG and improve its consumer acceptability.

### 2.2. Rheological Behavior of OGs

#### 2.2.1. Steady Shear Rheological Behavior of OGs and MOGs

OG containing 7% GML had higher apparent viscosity than that containing 5% GML (Figure 2A). This indicated that organogelator with a higher content produces a denser network structure in the oil phase [21,22]. Both of them showed shear-thinning behavior so were non-Newtonian fluids [23]. Three kinds of hydrocolloids with different surface charges were introduced into the OG matrix, which were positive CS, neutral KGM and negative SA. Figure 2B displays that KGM had higher viscosity than CS and SA, and therefore we set KGM content at a lower level in the OG systems. All the hydrocolloids showed shear-thinning behavior.

Overall, the viscosity profile of OG as a function of shear rate significantly changed after the incorporation of hydrocolloids (Figure 2C–E). Compared to original OG, the initial viscosity of MOGs was lower (<10 Pa·s) after mixing with hydrocolloids (Figure 2C–E). Then their viscosity sharply increased (>100 Pa·s) when the shear rate increased from 0.1 s^−1^ to 0.2 s^−1^. Subsequently, their viscosity decreased with the increase of shear rate from 0.2 s^−1^ to 100 s^−1^, which showed the shear-thinning behavior. This viscosity profile as a function of shear rate was independent on the type and concentration of hydrocolloids, which demonstrated that the formation of OG was still dominated by the self-assembly of GML. This phenomenon was similar to previous research on organogel–alginate hybrid microparticles, where the addition of organogels made the viscosity of the alginate solution have the trend of increasing first and then decreasing [18]. However, the maximum viscosity of KGMOG and CSOG increased with higher KGM and CS concentration, while that of SAOG kept almost constant, regardless of SA concentration. As mentioned above, GML has a low negative charge [24], and hence positive CS would interact with the OG matrix, while SA had repulsive interaction with oil droplets. Moreover, although the KGM solution had higher viscosity than CS, there was no apparent difference between the KGMOG and CSOG. The results indicated that the structure of the OG dominated the apparent viscosity after mixing with hydrocolloids. However, the shear-thinning behavior of the OGs could be regulated in the presence of hydrocolloid and the viscosity fluctuation of OGs was related to the surface charge of the hydrocolloid.

#### 2.2.2. Frequency-Dependent Rheological Behavior of MOGs

The viscoelastic property of the matrix can be characterized by determining the storage (G′) and loss (G″) modulus. It is viscous at G′ < G″, and elastic at G′ > G″ [25]. Figure 3A displays that elasticity dominated in OGs. G′ kept almost constant with increasing frequency and started to decrease at the high frequency of 100 Hz, while G″ slightly decreased and then significantly increased at the frequency of 5~100 Hz. The results indicated that OG was elastically stable, but gradually became viscous in the high frequency region, which was consistent with the change of viscosity (Figure 2A). As compared to the original OG, CSOGs and SAOGs showed a much lower modulus, and KGMOGs had the highest modulus, due to the higher viscosity of KGM. In general, the three MOGs exhibited frequency dependence of G′ and G″, reflecting that the interaction approach between GML and the hydrocolloid might be noncovalent “physical” breakable or deformable cross-links [26]. The modulus of all MOGs increased with higher hydrocolloid concentration, indicating the formation of a stronger network in the gel structure [22]. The effect of SA concentration was the lowest, which was consistent with that on the viscosity (Figure 2C–E). This might be because SA with a negative charge had a certain electrostatic repulsion with negatively charged GML, which was not conducive to the gel structure. Back to the viscoelasticity, G′ of OG was higher than G″ at the low frequency range. Then both of them increased with increasing frequency and G″ became higher than G′ after a certain frequency. As compared to the viscoelasticity of pure OGs, charged hydrocolloid decreased the mechanical stability of the OGs. The frequency at the intersection point of G′ and G″ shifted to a higher value with higher CS concentration, demonstrating that CS at higher concentration strengthened the stability of OGs. For SAOG, the intersection point of G′ and G″ shifted to a lower frequency at 1.5% SA. For KGMOG, G′ was always greater than G″ in all frequency ranges, which demonstrated that KGM addition would not destroy the stability of the OG [20].

Furthermore, the ratio G″/G′ (defined as tan δ) is more convenient to reflect the influence of hydrocolloids on the stability of OGs. Tan δ < 1, indicating that elasticity is dominant in the system, belongs to the gel. Tan δ > 1, indicating that the viscosity dominates the system, belongs to the fluid. Tan δ = 1, where the viscosity is equal to the elasticity, represents the sol-gel transition point [27]. The parameter of tan δ kept almost lower than 1 for OG and KGMOG (Figure 3A′,C′), demonstrated that the gel was relatively stable. Similar results also appeared in the regulation of KGM on the quality of low-fat processed cheese and mayonnaise [20,28]. The gel-sol transition occurred in CSOG and SAOG (Figure 3B′,D′), indicating that charged hydrocolloids might be much easier to affect the stability of OGs.

#### 2.2.3. Temperature-Dependent Rheological Behavior of MOGs

GML-induced OGs can be produced by the self-assembly of GML at low temperature. Hence, the influence of temperature on the modulus of OG was studied. G′ of original OG was higher than G″ at low temperature (3~30 °C, Figure 4A). At the beginning of heating, the gel system was a stable solid state. As temperature continually increased, modulus started to sharply decrease and G″ became higher than G′ at around 47 °C, which indicated that the systems turned into viscous systems. The gel structure was damaged at this high temperature, due to the melting point of GML [22]. It was reported that the system exhibited a viscous behavior at high temperature when the gelling agent was completely molten [29]. The addition of hydrocolloids maintained the decreasing profile of modulus with increasing temperature (Figure 4B–D). The transition point of gel to sol was similar, independent of the type of hydrocolloid. Modulus sharply decreased when the temperature was higher than 37 °C, indicating that the melting temperature of OG decreased around 10 °C. These results could be due to the melting of GML crystals (breakup of the OGs) and the hydrocolloid might affect the self-assembly behavior of GML at low temperatures. A similar result was also reported by Zheng et al., who found that κ-carrageenan hydrogel could reduce the crossover temperature of monoglyceride oleogels [30]. Gravelle et al. also found the crossover temperature of stearyl alcohol and stearic acid (SOSA) oleogels was reduced by adding ethylcellulose (EC), and attributed the altered the crystallization behavior of SOSA to the addition of EC [31]. The DSC curve also proved that the addition of hydrocolloid reduced the melting temperature of OGs (Appendix A). The original OG had the highest melting peak temperature (~37 °C), and the addition of hydrocolloid made the melting peak decrease about 6~7 °C, which was almost in accordance with that of the temperature sweep test (Figure 4). It should be noted that during the cooling process, the addition of hydrocolloid shifted the crystallization peak to a relative higher temperature, which increased about 2 °C. This result might be because the hydrocolloid network could act as a scaffold to accelerate GML crystallization, resulting in the increase of the crystallization temperature [31]. 

### 2.3. Microstructure of OGs

Before gelling, oil particles were uniformly distributed in the water phase (Figure 5). Aggregations of oil particles were observed in the presence of CS. The addition of KGM and SA caused the coacervation of oil particles, especially for SA. After cooling, the gel formed and the oil phase linked together. The accumulation and assembly of oil particles contributed to the grape-like gel structure of OG. In the presence of hydrocolloids, the gel network was more condensed and oil particles were difficult to distinguish. After melting and regelling, the gel structure could be recovered, demonstrating that all OGs were thermal-reversible, independent of the presence of hydrocolloids. All the gels in the presence of hydrocolloids were stable after freeze-drying and the microstructure was characterized by SEM (Figure 6). Due to the oily surface after drying, there were no images for original OG. KGMOG showed a relatively smooth surface, while CSOG had the roughest surface with lots of wrinkles. The cross-section structure of all the MOGs showed the three-dimensional network. CSOG was more compact and the cavities in KGMOG were larger. The cross-section of SAOG seemed to be fragile and like scrap paper, which might be closely related to the negative charge carried by SA, resulting in strong electrostatic repulsion with oil droplets and causing a relatively weak gel structure [32].

## 3. Materials and Methods

### 3.1. Materials

Glycerol monolaurate (GML) was obtained from Sinopharm Chemical Reagent Co., Ltd. (Shanghai, China). Medium chain triglyceride (MCT) was purchased from Wuhan Boxing Chemical Co. (Wuhan, China). Chitosan (CS, degree of deacetylation of 85%, viscosity of 1% chitosan solution at 20 °C at 1250 mPa·s, Mw: 2.78 × 105 Da by dynamic light scattering) was provided by Zhejiang Golden-Shell Pharmaceutical Co., Ltd. (Zhejiang, China). Konjac Glucomannan (KGM, food grade, purity ≥ 95%) was purchased from Qiangsen konjac Products Co., Ltd. (Wuhan, China). Sodium alginate (SA, LOT24686) was provided by TIC GUMMS (Guangzhou, China). Nile red was acquired from Sigma-Aldrich (Shanghai, China). All the other chemicals were of analytical grade.

### 3.2. Methods

#### 3.2.1. Preparation of OGs

OGs were prepared by mixing GML and MCT. Specifically, a certain amount of GML was added into MCT. The mixture was then stirred under 60 °C until the dissolution of GML. The amount of the mixture was 4 g. GML level was set at 5% and 7%. After dissolution, the mixture was place in the refrigerator (4 °C) for the formation of OG.

#### 3.2.2. Preparation of Modified OGs (MOGs)

Preparation of hydrocolloid solutions: a certain amount of CS powder was added into 1% acetic acid with overnight stirring for the dissolution of CS. The obtained CS solution was centrifuged at 8000 rpm for 10 min to remove any impurities. By the same method, KGM and SA solutions were prepared by dissolving them in distilled water. CS and SA concentrations varied from 0.75%, 1% to 1.5%. KGM concentration was set at 0.2%, 0.3% and 0.4%.

Preparation of oil phase: the method was the same as above in Section 3.2.1., with modification, 1.68 g of GML was mixed with 2.32 g of MCT. The mixture was then stirred under 60 °C until the dissolution of GML.

Formation of MOG: under 60 °C, the mixture of GML and MCT (4 g) was slowly added into hydrocolloid solution (20 g) with stirring. After emulsification, the resultant mixture was stirred in an ice bath to make it cool. When the mixture became more viscous, it was put in the refrigerator to complete the gelation process. After mixing with CS, KGM and SA solution, the obtained MOGs were named as CSOG, KGMOG and SAOG, respectively. In the MOGs, MCT and GML content was 11.6% and 7%.

### 3.3. Oil holding Capacity and Gelling Time

Oil holding capacity: firstly, about 1 mL of the solution sample was added to the micro-centrifuge tube and the centrifuge tube was placed in the refrigerator at 4 °C for 1 h to form a gel. Next, the centrifuge tube was centrifuged (9167 g, 2 °C) for 15 min. Finally, the centrifuge tube was inverted for 3 min, and the excess precipitation was absorbed. The centrifuge tube, before and after filling with the sample, was weighed for the calculation.
(1)$$\%\mathrm{OBC}=100-\frac{\left(b-a\right)-\left(c-a\right)}{b-a}\times 100\%$$
where *a* is the mass of empty micro centrifuge tube, *b* is the mass of the filled centrifuge tube after forming the gel, *c* is the mass of the filled centrifuge tube after absorbing the excess precipitation.

Gelling time: after the polymer was mixed with the molten OG, the mixture was stirred up in the ice bath for thorough mixing. The gelling time was recorded by stopwatch since OGs and MOGs were placed in the refrigerator at 4 °C until gel formed.

### 3.4. Texture Characterization

The physical properties of OGs and MOGs were characterized using a TA.XT. PLUS physical analyzer (STABLE MICRO.SYS, Godalming, UK), P/45C probe. The texture characteristics of different OGs were determined and recorded by firmness, stickiness and adhesiveness. All measurements were carried out in three times.

### 3.5. Rheological Properties of OGs

The steady and dynamic rheological properties of the OGs were determined using an AR-2000ex rheometer (TA Instruments, West Sussex, UK). The measurements were carried out in a parallel-plate geometry system and the measuring gap was set at 1000 μm. The sample was deposited on the plate and waited for 5 min to allow temperature equilibration prior to measurement. For the steady-state flow measurements, the shear rate was set from 0.1 to 100 s^−1^ and the apparent viscosity was obtained from TA data analysis software (TRIOS software 4.4.0.41128, Saugus, MA, USA). All dynamic tests were performed within the linear viscoelastic region. The frequency was oscillated from 0.1–100 rad/s with the strain of 1%. The conditions of the frequency sweep test were 0.1–100 Hz and the selection of stress value was 0.1%. The relationship between the elastic modulus (G′) and the loss modulus (G″) versus frequency was recorded. All the above measurements were performed at 25 °C.

To study the thermal-sensitivity of OGs, the change of G′ and G″ was tested as a function of temperature. The temperature altered from 5–50 °C and the heating rate was 2 °C/min. The temperature control system adopted Haake Universal temperature control device (Thermo Scientific, Tewksbury, MA, USA) and the accuracy was ±0.01 °C. All measurements were carried out three times.

### 3.6. Microstructure Characterization

#### 3.6.1. Fluorescent Microscopy

The microstructure of the OGs was characterized using an ECLIPSE 80i positive fluorescence microscope (NIKON, Tokyo, Japan). Nile red (0.01 wt.% in ethanol) was used to dye the oil phase. The samples were analyzed under molten and gelled state.

#### 3.6.2. Scanning Electronic Microscopy

The OGs were placed in a plastic cup and quickly frozen by adding liquid nitrogen and were then kept at −18 °C overnight to ensure thorough freezing, followed by lyophilization for 48 h. The surface and cross section of the dried OGs were observed by scanning electron microscopy (JSM-6390LV, NTC, and Tokyo, Japan) at a voltage of 30 keV.

### 3.7. Statistical Analysis

All experiments were performed in triplicate on freshly prepared samples. The results were then reported as averages and standard deviations of these measurements. Samples were considered to be statistically significant if *p* < 0.05.

## 4. Conclusions

In summary, OG was prepared by the self-assembly of GML at low temperature. Then the structure and properties of the OG were modulated by the incorporation of hydrocolloids. The OG had high viscosity and then showed shear-thinning behavior. MOGs had much lower viscosity than OG after mixing with hydrocolloids. Subsequently, the viscosity of MOGs sharply increased under the low shearing force, and then decreased with a higher shear rate. The fluctuation of the MOGs’ viscosity as a function of hydrocolloid content was related to the charge property of the hydrocolloid. Dynamic rheology study showed that original OG and KGMOG were stable against the change of frequency, while CSOG and SAOG exhibited frequency dependence of G′ and G″, reflecting that the regulation method might be a noncovalent interaction. The gelation temperature of gels could be decreased around 10 °C and the gelling time was shortened. OG in the presence of hydrocolloids had a more compact network structure, causing a stronger oil holding capacity. Overall, a charged hydrocolloid had a more significant effect than a neutral one. These results may facilitate the research of property modulation on organogels, and the development of functional foods like novel nutrient carriers or low-fat margarine alternatives.

## Figures and Tables

**Figure 1 molecules-25-05117-f001:**
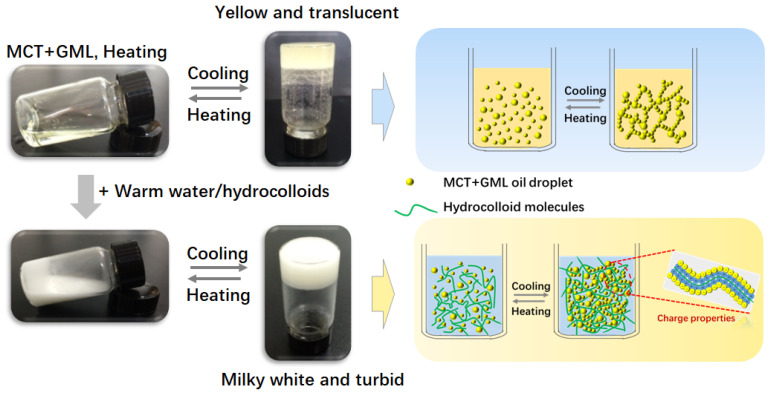
Appearance and conversion of OGs in the absence and presence of hydrocolloids. The addition of water or hydrocolloid solution caused the milky turbidity. The added hydrocolloids interacted with oil droplets, depending on the surface charge of the hydrocolloids.

**Figure 2 molecules-25-05117-f002:**
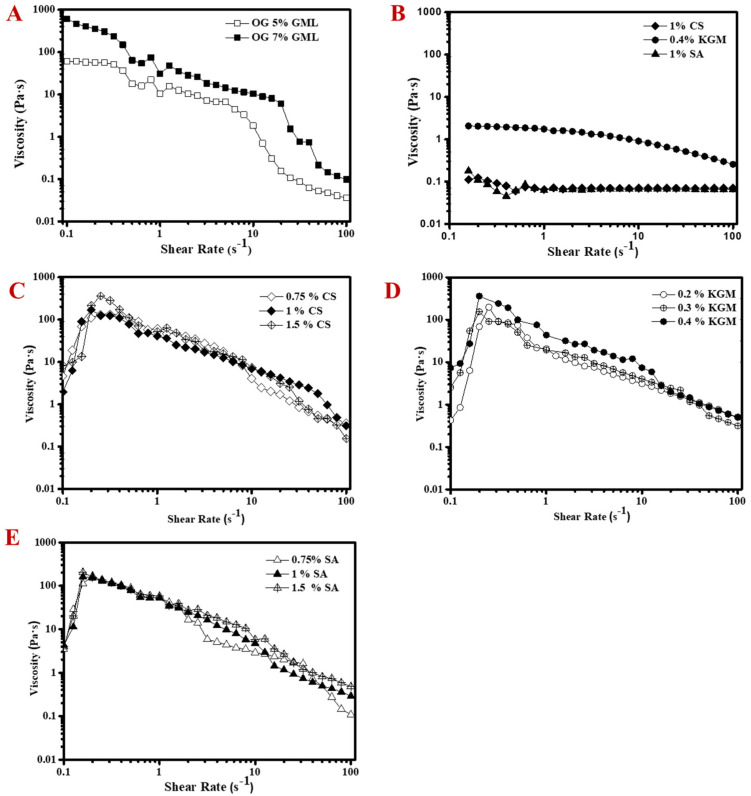
Viscosity profile as a function of shear rate: (**A**) influence of GML content on pure OGs, (**B**) viscosity of hydrocolloids (1% CS, 0.4% KGM and 1% SA), influence of (**C**) CS concentration (0.2%, 0.3% and 0.4%), (**D**) KGM concentration (0.75%, 1% and 1.5%), and (**E**) SA concentration (0.75%, 1% and 1.5%) on the viscosity of MOGs.

**Figure 3 molecules-25-05117-f003:**
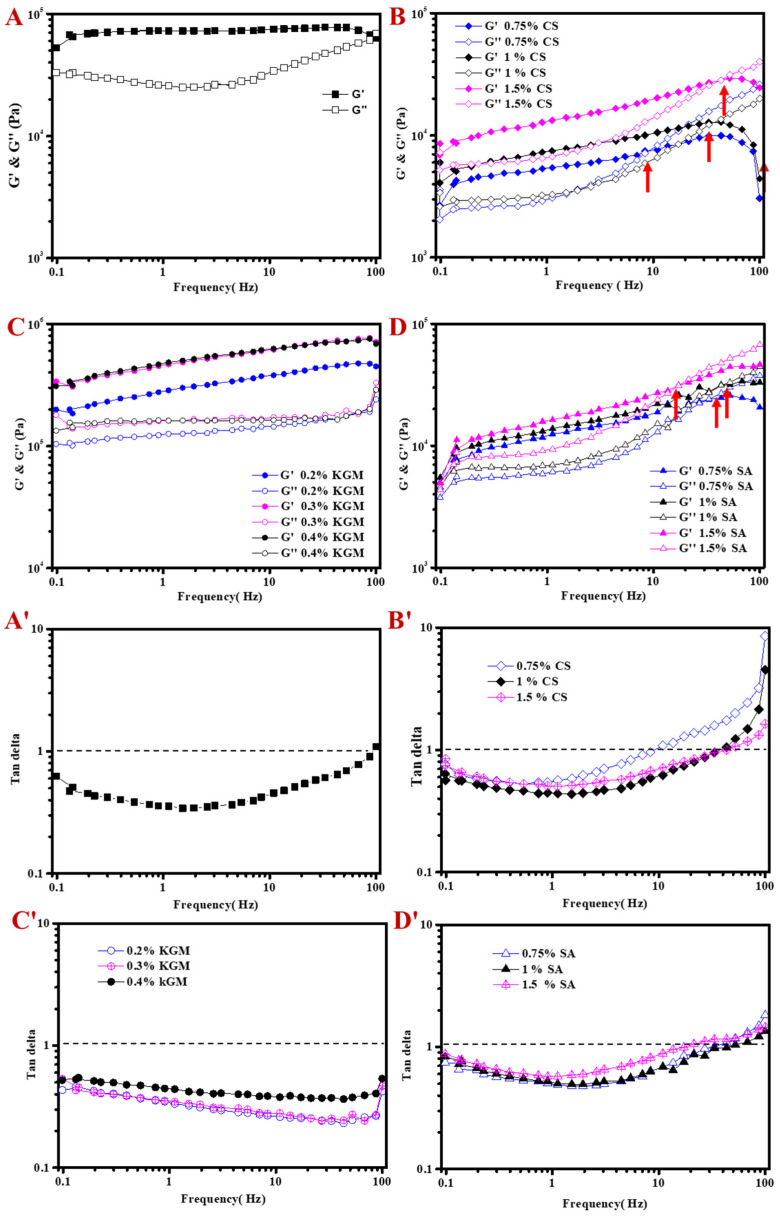
Storage modulus (G′), loss modulus (G″) and tan delta as function of frequency, (**A** and **A′**) OG, (**B** and **B′**) CSOG, (**C** and **C′**) KGMOG and (**D** and **D′**) SAOG.

**Figure 4 molecules-25-05117-f004:**
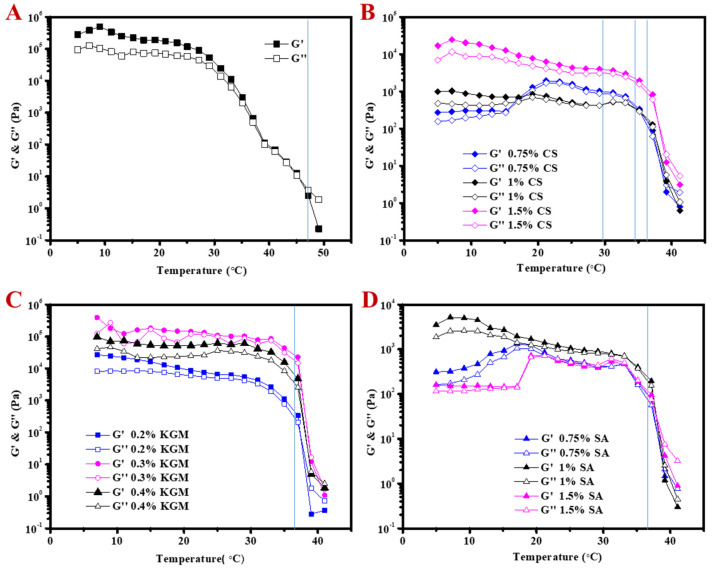
Storage modulus (G′) and loss modulus (G″) as a function of temperature, (**A**) OG, (**B**) CSOG, (**C**) KGMOG and (**D**) SAOG.

**Figure 5 molecules-25-05117-f005:**
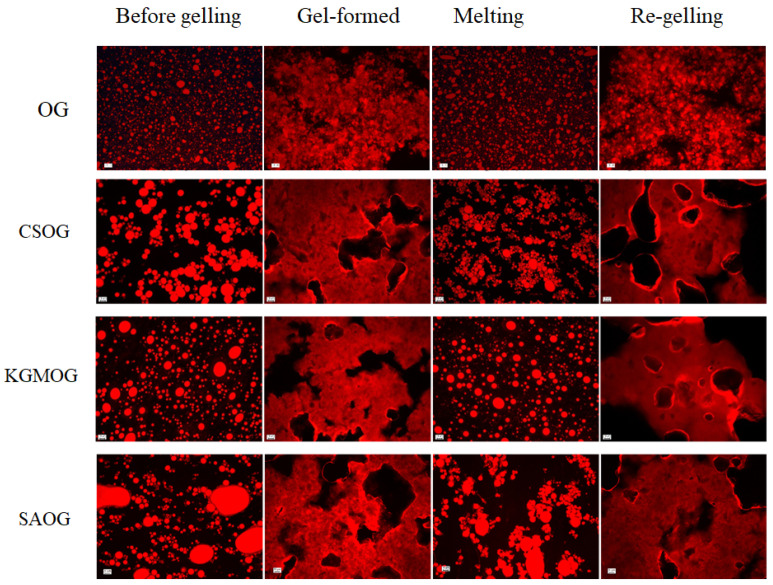
Fluorescent microscopic images of OGs and MOGs in the presence of hydrocolloid. OG is the original organogel, CSOG is OG + 1% CS, KGMOG is OG + 0.3% KGM, and SAOG is OG + 1% SA. The scale bar is 10 μm.

**Figure 6 molecules-25-05117-f006:**
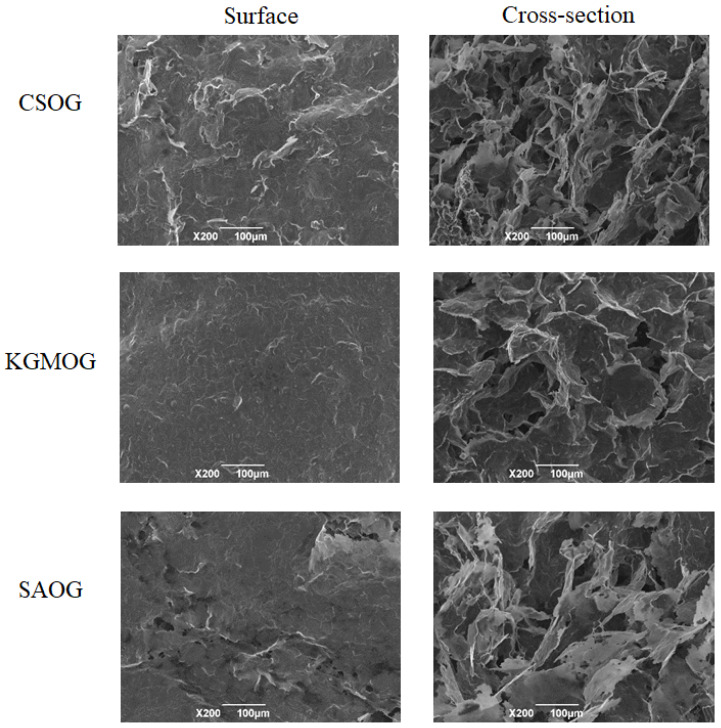
SEM images of MOGs. CSOG is OG + 1% CS, KGMOG is OG + 0.3% KGM, and SAOG is OG + 1% SA. The scale bar is 100 μm.

**Table 1 molecules-25-05117-t001:** Gelling time, centrifuge stability and textural properties of organogels (OGs) and MOGs in the presence of hydrocolloids, CSOG is OG + 1% CS, KGMOG is OG + 0.3% KGM, and SAOG is OG + 1% SA. (Statistical analysis were carried out for the same parameter and different samples. Samples designated with different letters (a, b, c and d) indicate significantly difference, *p* < 0.05).

Samples	Gelation Time	Stability (%OBC)	Separation	Texture Characteristics		
				Firmness (g)	Stickiness (g)	Adhesiveness (g·sec)
OG	21 min	65.3%	Obvious separation	213.04 ± 22.88 ^a^	−165.03 ± 25.53 ^c^	−129.80 ± 20.91 ^a^
CSOG	1 min 42 s	93.9%	No separation	470.25 ± 40.96 ^d^	−333.55 ± 84.47 ^a^	−93.74 ± 11.35 ^a^
KGMOG	1 min 48 s	99.2%	No separation	377.02 ± 49.37 ^c^	−264.31 ± 57.19 ^ab^	−121.59 ± 51.54 ^a^
SAOG	1 min 40 s	84.1%	Slight separation	297.79 ± 40.17 ^b^	−227.33 ± 41.78 ^bc^	−105.54 ± 25.93 ^a^

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
