# Peer review of "Structure and Rheological Properties of Glycerol Monolaurate-Induced Organogels: Influence of Hydrocolloids with Different Surface Charge"

_molecules, 2020, doi:10.3390/molecules25215117_

Round 1
Reviewer 1 Report
The authors reported the structure and rheological properties of glycerol monolaurate induced organogels. They believe that the hardness, stickiness and the oil holding stability of OGs could be modified by introducing hydrocolloids. Some of the findings in this work may significant for preparing food without trans- fatty acids, but some concerns must be addressed before publication. I recommend its publication in Molecules after major revision:
- Inthe Abstract, the author should provide valid information rather than fuzzy introduction. For example, the author said “viscosity of hydrocolloid incorporated-OG increased at low shear rate and then decreased at high shear rate” and “ hydrocolloids improved the hardness, stickiness and the oil holding stability of OGs”. The specific data weren’t given in abstract.
- In the Introduction, the authorintroduced the research progress of OG, but I suggest the authors to discuss the related to the topic of this article rather than something irrelevant discussion.
- The functionof Fig. 1? I can’t understand your means. The author should used a specific schematic illustration to present the preparation process or mechanism rather than simple photos. Moreover, I don't see the difference from above photos given.
- The format of Table 1 should be modified.
- Istrongly suggest the authors to improve the quality of figures, and some figures (e.g. Fig. 2, 3 and 4) are difficult to see clearly. Thus, some significant results can be confirm due to the unclear figures.
- In Fig. 5. What is the large agglomerations in fluorescent microscopic images of SAOG.
- The authorrepeated emphasis that their OG was edible, but I can’t found any data to prove their statement. I suggest the authors to do some characterization to improve the quality of this work.
Author Response
- In the Abstract, the author should provide valid information rather than fuzzy introduction. For example, the author said “viscosity of hydrocolloid incorporated-OG increased at low shear rate and then decreased at high shear rate” and “hydrocolloids improved the hardness, stickiness and the oil holding stability of OGs”. The specific data weren’t given in abstract.
-As suggested by reviewer, we modified the abstract and some specific results were presented. The revised part was displayed as follow.
“Rheological studies demonstrated that OG and pure hydrocolloid solution showed shear-thinning behavior. After incorporation of hydrocolloid, the initial viscosity of OG was lowered from ~100 Pa·s to <10 Pa·s, and then the viscosity increased to more than 100 Pa·s at low shear rate of 0.1-0.2 s-1, which subsequently decreased with higher shearing rate. OG in the presence of hydrocolloids still kept the thermo-sensitivity, while the melting point of OG decreased with the incorporation of hydrocolloids. Hydrocolloids greatly shortened the gelling time of OG from 21 min to less than 2 min. The presence of hydrocolloids increased the particle size of oil droplets in the molten OG. Some aggregation and coacervate of oil droplets occurred in the presence of positive-charged CS and negative-charged SA, respectively. After gelling, the gel structure converted into a biphasic-like network. Hydrocolloids improved the hardness, stickiness and the oil holding stability of OGs increased by 18.8~33.9%.”
- In the Introduction, the author introduced the research progress of OG, but I suggest the authors to discuss the related to the topic of this article rather than something irrelevant discussion.
-As suggested by reviewer, we revised the introduction. Especially, we deleted the first paragraph. Up to now, the modification of OGs focused on selection of oil phase or gelators. Some works are about the preparation techniques. But the work about OGs and hydrophilic polysaccharides is lack. Hence, to improve the biocompatibility of OGs with food systems, we tried to combine polysaccharides and OGs.
- The function of Fig. 1? I can’t understand your means. The author should use a specific schematic illustration to present the preparation process or mechanism rather than simple photos. Moreover, I don't see the difference from above photos given.
-As suggested by reviewer, we re-organized the content of Fig.1. We presented the appearance of OGs in the absence and presence of hydrocolloids. Also their formation was provided.
- The format of Table 1 should be modified.
-As suggested by reviewer, we revised the format of Table 1 to make it clearer.
- I strongly suggest the authors to improve the quality of figures, and some figures (e.g. Fig. 2, 3 and 4) are difficult to see clearly. Thus, some significant results can be confirmed due to the unclear figures.
-As suggested by reviewer, we re-organized the figures and enlarged them.
- In Fig. 5. What is the large agglomerations in fluorescent microscopic images of SAOG.
-In response to reviewer’s comments, we think that those large agglomerations are the large oil droplets, due to the coalescence of oil droplets in the presence of SA.
- The author repeated emphasis that their OG was edible, but I can’t find any data to prove their statement. I suggest the authors to do some characterization to improve the quality of this work.
-In response to reviewer’s comments, edible OGs are commonly mentioned in the references. They are recognized by selecting the edible materials to produce them. I agree that some experiments should be carried out to evaluate the edible property of OGs. We think it is a good idea for our future work. However, the influence of hydrocolloid was emphasized in the present work.

Reviewer 2 Report
In this Manuscript, the authors produced Organogel (OG) by glycerol monolaurate as organogelators and demonstrated that OGs containing hydrocolloids are suitable for the food formula that can shear stress in the food processing stage. The findings of this manuscript could be useful for food processing company. However, a minor revision is required for getting the acceptance of this manuscript (see comments below).
Fig. 1 is not clearly explained. It is very hard to understand the results that showed by this figure. The main text and figure legend are required to expand for better understanding.
Similarly, the legend of Fig. 2 should be expended.
Author Response
Fig. 1 is not clearly explained. It is very hard to understand the results that showed by this figure. The main text and figure legend are required to expand for better understanding.
-As suggested by reviewer, we re-organized the content and revised the caption of Fig.1. We presented the appearance of OGs in the absence and presence of hydrocolloids. Also their formation was provided.
Figure 1 Appearance and conversion of OGs in the absence and presence of hydrocolloids. The addition of water or hydrocolloid solution would cause the milky turbidity. The added hydrocolloids would interact with oil droplets, depending on the surface charge of hydrocolloids.
Similarly, the legend of Fig. 2 should be expended.
-As suggested by reviewer, we revised the legend of Fig. 2.
‘Figure 2 Viscosity profile as a function of shear rate, (A) Influence of GML content on pure OGs, (B) Viscosity of hydrocolloids (1% CS, 0.4% KGM and 1% SA), (C-E) Influence of CS concentration (0.2%, 0.3% and 0.4%), KGM concentration (0.75%, 1% and 1.5%), and SA concentration (0.75%, 1% and 1.5%) on viscosity of MOGs.’
